# The Subcutaneous Administration of Beta-Lactams: A Case Report and Literary Review—To Do Small Things in a Great Way

Gabriele Maria Leanza [1], Beatrice Liguoro [1], Simone Giuliano [2,*], Chiara Moreal [3], Luca Montanari [3], Jacopo Angelini [4], Tommaso Cai [5,6], Rita Murri [1,7] and Carlo Tascini [2,3]

1   Dipartimento di Sicurezza e Bioetica, Università Cattolica del S. Cuore, 00168 Rome, Italy; gabrielemaria.leanza01@icatt.it (G.M.L.); beatrice.liguoro@guest.policlinicogemelli.it (B.L.); rita.murri@policlinicogemelli.it (R.M.)
2   Infectious Diseases Clinic, Azienda Sanitaria Universitaria del Friuli Centrale (ASUFC), 33100 Udine, Italy; c.tascini@gmail.com
3   Infectious Diseases Clinic, Department of Medicine (DAME), University of Udine, 33100 Udine, Italy; moreal.chiara@spes.uniud.it (C.M.); montolly1989@gmail.com (L.M.)
4   Pharmacology Institute, Azienda Sanitaria Universitaria del Friuli Centrale (ASUFC), 33100 Udine, Italy; jacopo.angelini@asufc.fvg.sanita.it
5   Department of Urology, Santa Chiara Regional Hospital, 38123 Trento, Italy; ktommy@libero.it
6   Institute of Clinical Medicine, University of Oslo, 0315 Oslo, Norway
7   Dipartimento di Scienze di Laboratorio e Infettivologiche, Fondazione Policlinico Universitario A. Gemelli IRCCS, 00168 Rome, Italy
*   Correspondence: simone.giuliano@asufc.sanita.fvg.it

**Abstract:** The subcutaneous (s.c.) route is a commonly used method for delivering various drugs, although its application in the administration of antibiotics is relatively uncommon. In this case, we report a successful treatment of nosocomial pneumonia using piperacillin/tazobactam via continuous subcutaneous administration. Furthermore, this article provides an overview of the current literature regarding the s.c. administration of beta-lactam antibiotics. Based on our analysis, we identified only 15 studies that described the s.c. use of beta-lactam antibiotics in human subjects. Among these studies, cephalosporins were the most extensively investigated antibiotic class, with 10 available studies. According to the study findings, all three antibiotic classes (cephalosporins, penicillins, and carbapenems) demonstrated a similar pharmacokinetic profile when administered via the subcutaneous route. The subcutaneous route appears to be associated with a lower peak serum concentration ($C_{max}$) but a comparable minimum blood concentration ($C_{min}$) and an extended half-life ($t_{1/2}$) when compared to conventional routes of antibiotic administration. Further research is necessary to determine whether subcutaneously administered beta-lactam antibiotics in human subjects achieve pharmacodynamic targets and demonstrate clinical efficacy.

**Keywords:** subcutaneous administration; beta-lactams; PK/PD

## 1. Introduction

Bacterial infections are a major cause of morbidity and mortality in the general population, posing many challenges for the clinician [1]; one of these challenges is choosing the appropriate route for the administration antibiotics. The most commonly used routes for the administration of antibiotics include intravenous (i.v.), oral, and intramuscular (i.m.), each with its own set of advantages and disadvantages. The majority of hospitalized patients receive antibiotic therapy with i.v. administration, using both continuous and intermittent infusion methods. However, i.v. antibiotic administration through venous catheter can sometimes be difficult or contraindicated. The i.m. route may provide an alternative, but it is frequently contraindicated due to the prevalent use of anticoagulant therapy, particularly among geriatric and bedridden patients, and is associated with delayed and erratic drug absorption [2–4]. S.c. administration offers a potential solution to

these complications, particularly in patients with poor venous access and contraindications to multiple intra-muscle injections. Moreover, it could serve as a viable option for home treatment, reducing hospitalization days and the associated costs of extended stays. In this context, the primary objective of this study is to present the case of a patient with nosocomial pneumonia treated with piperacillin/tazobactam via subcutaneous administration after a few days of i.v. therapy. Additionally, we conducted a comprehensive literature review to gather the current evidence regarding the utilization of beta-lactams via the subcutaneous route.

## 2. Case Report

We present the case of an 88-year-old patient who was admitted to the Infectious Disease clinic at University Hospital in Udine due to the incidental detection of a positive SARS-CoV-2 nasal-pharyngeal swab result during hospitalization for head injury and a stable C2 fracture without surgical indication. At day 30 of hospitalization, the patient exhibited a decrease in oxygen saturation and went through an episode of fever (38.7 °C). Vital signs were stable, and the laboratory results reflected a C-reactive protein (CRP) elevation (82 mg/L). In the context of this clinical deterioration, a chest radiograph was performed, revealing a widespread and bilateral accentuation, blurring of the vasculo-interstitial pattern, and subtle parenchymal consolidations in the upper and lower right perihilar regions. Due to suspected hospital-acquired pneumonia, an empiric antibiotic therapy with piperacillin/tazobactam continuous infusion (4.5 g loading dose and then 18 g maintenance dose in 24 h continuous infusion) was started. The blood cultures collected before the start of antibiotic therapy were negative. After 2 days, we decided to switch the administration route of piperacillin/tazobactam from i.v. to s.c. due to the unavailability of suitable veins for injection. After the insertion of Venflon 20 Gauge catheter for s.c. infusion in the thigh (Figure 1), we initiated s.c. infusion of 4.5 g of piperacillin/tazobactam in 50 mL saline solution through a normal i.v. line without an elastomeric or an infusion pump. Every 6 h, we temporarily discontinued the infusion, massaged the injection site, and then reattached the infusion, changing its location to the other thigh every 24 h. Through this administration schedule, we aimed to simulate a continuous infusion at a rate of 8.4 mL/h. During the following days, therapeutic drug monitoring (TDM) was conducted, revealing a blood concentration of piperacillin of 52.39 mg/L after 4 days and 93.73 mg/L after 8 days following the switch from i.v. to s.c infusion. The patient went through a clinical improvement and a reduction in the inflammation markers (10 days after the start of the therapy, the CRP was 9.45 mg/L). For this reason, the treatment was discontinued after a duration of 10 days. The piperacillin blood concentration measured at the end of antibiotic therapy was 156.39 mg/L.

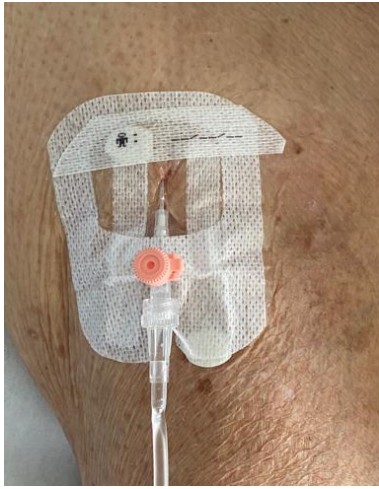

**Figure 1.** Venflon 20 Gauge catheter used for piperacillin/tazobactam s.c. infusion (picture taken at the end of the infusion).

No adverse events, whether systemic or local to the injection site, were observed.

## 3. Methods

### 3.1. Search Strategy and Article Identification

Published articles (until August 2023) assessing the administration of beta-lactam antibiotics via the s.c. route in humans were identified through computerized literature searches using MEDLINE (National Library of Medicine Bethesda MD), Scopus, and Web of Science by reviewing the references of retrieved articles. The search strings used for bibliographic research are listed in the Supplementary Materials. English and Italian language restriction was applied. No age-range restriction was applied.

### 3.2. Eligibility Criteria

Studies of any design excluding abstract form, letter to editor, oral presentation, reviews, guidelines, and clinical trial protocols, which reported data on the administration of beta-lactam antibiotics subcutaneously in humans, were eligible for inclusion in our review.

### 3.3. Study Selection and Data Extraction

An eligibility assessment was performed independently by two investigators (G.M.L. and B.L.). Each investigator was blinded to the other investigator's eligibility assessment. In case of disagreement between the two reviewers, a third reviewer (C.T.) was consulted. Data from each study were verified for consistency and accuracy.

## 4. Results

Our literature review found 15 studies describing the s.c. use of beta-lactams in human subjects. As shown in Table 1, among these studies, we found eight pharmacokinetic (PK) studies, three observational studies, two case reports, and two case series. The most extensively studied antibiotic class for s.c. administration in humans is cephalosporins, with ten studies identified. We found four studies on humans describing the carbapenem administration via the s.c. route and three studies describing the use of penicillins via s.c. route.

Evidence regarding the clinical and pharmacokinetic efficacy of s.c. administration for all three classes of antibiotics is presented in the following sections. The PK measures documented in the included studies are reported in Table 2.

### 4.1. Cephalosporins

Our literature review found 4 PK studies on healthy volunteers regarding the s.c. administration of ceftriaxone (2 studies), cefepime, and ceftazidime. All these studies demonstrated that the s.c. administration of these three drugs resulted in lower and delayed peak blood concentrations ($C_{max}$) compared to the i.v. infusion. However the area under the concentration–time curve (AUC), the minimum blood concentration ($C_{min}$) and the $t_{1/2}$ was comparable between the two different administration routes [5–8].

Some authors attempted to enhance the absorption of cephalosporins through s.c. administration by using some agents that increase connective tissue permeability, such as hyaluronidase, or the subcutaneous blood flow rate (such as mentholated warm compresses) [6,7]. These strategies appeared to exert a favorable influence on cephalosporin $C_{max}$ and $T_{max}$ when administered subcutaneously.

The studies and case reports examining clinical outcomes showed a satisfying efficacy for subcutaneously administered cephalosporins. For example, Pilmis et al. described a cohort of 12 patients, 10 with bone and joint infections and 2 with bloodstream infections secondary to urinary tract infections, all treated with cefepime. All 12 patients achieved a positive clinical outcome, with no relapses of the infection at 6 months [9]. In 2014, Gauthier et al. conducted a comparative study on the use of s.c. and intramuscular (i.m.) ceftriaxone for the treatment of various types of infections in a cohort of 148 geriatric patients, with 38 patients receiving s.c. administration and 110 patients receiving i.m. administration.

The clinical cure rate in the s.c. group was only marginally lower than that observed in the i.m. group (70.3% vs. 76.2%) [10].

Local reactions to the s.c. injection (pain/itching/erithema, etc.) were the most common adverse events reported in the studies included in the review [8]. Pouderoux et al. reported a case of skin necrosis after rapid (a few seconds) s.c. ceftriaxone injection [11].

### 4.2. Carbapenems

Ertapenem is the only carbapenem studied in humans for s.c. administration. According to the study of Frasca et al., s.c. injection is associated with a much lower $C_{max}$ compared to i.v. administration. The AUC0-24s.c./AUC0-24i.v. in this study, however, was 0.99. In addition, when administered subcutaneously, ertapenem exhibited an intriguing prolongation of its $t_{1/2}$ [12].

The encouraging data regarding the efficacy of the s.c. administration of ertapenem have been confirmed by the PK study of Goutelle on a cohort of patient with bone and joint infections (BJI). The Monte Carlo simulation performed in this study identified a higher probability of target attainment (PTA) with the s.c. administration of ertapenem compared to the i.v. route [13].

In a case series published in 2021, Goutelle et al. documented a case of ertapenem s.c. treatment failure in a patient undergoing suppressive therapy for a prosthetic joint infection (PJI) caused by *E. cloacae*. This patient was treated with a thrice-weekly regimen of s.c. ertapenem (1 g on Monday and Wednesday and 2 g on Friday) and, after 7 months, expereinced a relapse. In this case, the issue was the lack of knowledge regarding the ertapenem minimum inhibitory concentration (MIC) for *E. cloacae* at the onset of treatment. Subsequently, it was discovered that the MIC value was 0.38 mg/L, and the PK profile revealed that the thrice-weekly regimen did not achieve a sufficient fraction of time above the MIC (fT > MIC) [14].

In the study by Roubaud-Boudron, it was documented that the ertapenem PTA (fT > MIC > 40%) was achieved in 93.60% of the patients treated with ertapenem s.c. infected by a microorganism displaying an ertapenem MIC of up to 1 mg/L compared to a PTA of 86.20% in the group of patients treated with ertapenem i.v. (considering the same MIC range). In this cohort, the 15-day survival percentage was 75% in the s.c. group and 70% in the i.v. group [15].

### 4.3. Penicillins

We found three PK studies examining the s.c. administration of penicillin in human subjects. The PK data extracted from these studies exhibit similarities to those observed in the two previously mentioned beta-lactams. These studies provide evidence of a lower and delayed $C_{max}$, and a higher $C_{min}$ and a longer $T_{1/2}$, associated with the s.c. administration, in contrast to the comparator group receiving i.v. or i.m. administration [16–18].

Kado et al. conducted a prospective pharmacokinetic study involving 15 healthy volunteers who received 1.2 million IU benzathine penicillin through either i.m. or s.c. administration (seven patients in the s.c. group vs. eight patients in the i.m. group). Simulations of penicillin concentration–time profiles revealed that both s.c. and i.m. administration resulted in blood concentration levels exceeding 20 ng/mL for approximately 30% of a 28-day period. However, when targeting a concentration of 10 ng/mL, the s.c. administration of 1.2 million IU consistently achieved the desired blood concentration throughout the entire interval between injections (100% vs. 65% for i.m. administration) [17].

**Table 1.** Studies included in the review.

| Author, Publication Year | Study Design | Drug, Dosage | Patients (s.c. vs. Control Group) | Characteristics | PK Measures (s.c. vs. Control Group) | Clinical Outcome |
|---|---|---|---|---|---|---|
| Pilmis, 2020 [9] | Retrospective observational study | Cefepime, 3.1 g/day | 12 s.c. vs. 12 i.v. | 10 BJI, 2 BSI at urinary departure | • $C_{min}$ (median): 29.1 mg/L vs. 31.9 mg/L | no recurrence of infection at 6 months |
| Walker, 2005 [8] | PK study | Cefepime, 1 g | 10 (s.c. vs. i.v.) | Healthy volunteers | • $C_{max}$ (mean): 36.1 mg/L vs. 29.6 mg/L<br>• $T_{1/2}$ (mean): 2.34 h vs. 2.37 h<br>• $AUC_{0-\infty}$ (mean): 134.8 h·mg/L vs. 137 h·mg/L | - |
| Borner, 1985 [5] | PK study | Ceftriaxone (0.5 g s.c. vs. 0.5 g i.v. vs. 2 g i.v) | 10 | Healthy volunteers | • $C_{max}$ (mean): 37.1 mg/L vs. 83.8 mg/L<br>• $C_{min}$ (mean) at 24 h: 6.6 mg/L vs. 6.5 mg/L<br>• $T_{max}$: 138′ vs. 37.4′<br>• AUCsc/AUCev: 0.96<br>• $V_d$ (mean): 8.3 L vs. 11.5 L | - |
| Harb, 2010 [7] | Phase I trial single-blind | Ceftriaxone, 1 g | 29 (ialuronidase-facilated s.c. infusion vs. standard s.c. infusion vs. i.v. infusion) | Healthy vounteers | • $C_{max}$ (mean): 92 μg/mL vs. 82.2 μg/mL vs. 150 μg/mL<br>• $T_{max}$ (mean): 2.02 h vs. 3.02 h vs. 0.502 h<br>• $T_{1/2}$ (mean): 7.97 vs. 8.28 h vs. 8.25 h<br>• $AUC_{0-t}$ (mean): 1139.3 μg·h/L vs. 1115.6 μg·h/L vs. 1065 μg·h/L | - |
| Gauthier, 2014 [10] | Retrospective obseervational study | Ceftriaxone, | 38 s.c. vs. 110 i.m. | Patients with >75 years (15 pneumonia, 12 UTI, 3 AI, 3 colecistitis, 1 IE, 1 ABSSI) | Not specified | Clinical cure: 70.3% vs. 76.2% |
| Michelon, 2019 [19] | Case report | Ceftazidime, 1 g | 1 | *Pseudomonas aerouginosa* catheter related-UTI | Not specified | Clinical and laboratoristic cure |
| Duron, 2019 [20] | Case report | Ceftazidime, 6 g/day | 1 | VAP with BAL isolation of P. aeruginosa MSSA and Achrobacter xylosoxidans | Not specifiend | Clinical and laboratoristic cure |
| Ebihara, 2016 [6] | PK study | Ceftazidime, 0.5 g/10 mL in continuous infusion (without mentholate compression vs. mentholate compression) | 1 | Healthy volunteer | • $C_{max}$: 44.8 μg/mL vs. 57.4 μg/mL<br>• $T_{max}$: 205′ vs. 201′ | - |

**Table 1.** *Cont.*

| Author, Publication Year | Study Design | Drug, Dosage | Patients (s.c. vs. Control Group) | Characteristics | PK Measures (s.c. vs. Control Group) | Clinical Outcome |
|---|---|---|---|---|---|---|
| Pouderoux, 2019 [11] | Case series | Ceftriaxone, 1 g/24 h; ceftazidime 2 g/24 h; ertapenem 1 g/24 or 1 g/12 h | 10 (ceftriaxone 2, ceftazidime 1, ertapenem 6, 1 patient received ceftriaxone for 8 days then switched to ertapenem) | 7 PJI, 3 CO | For patients treated with ertapenem and ceftazidime $C_{min}$ was always higher than the targeted pathogen MIC | 6/10 had a clinical success, 2/10 presented a clinical failure after the s.c. antibiotic therapy, 1 presented a clinical failure before the s.c. antibiotic therapy, 1 lost to follow-up |
| Goutelle, 2021 [14] | Case series | Ceftazidime, ertapenem, ceftriaxone | 10 (ceftazidime 4, ertapenem 4, ceftriaxone 2) | BJI in chronic suppressive therapy | 2 cases with $f$T/MIC < 50% (1 patient treated with ceftazidime and 1 patient with ertapenem) | 9/10 have not developed a recurrence of infection |
| Frasca, 2009 [12] | PK study | Ertapenem, 1 g | 6 (s.c. vs. i.v.) | 5 VAP, 1 surgical wound infection | <ul><li>$C_{max}$ (mean): 43 µg/mL vs. 115 µg/mL</li><li>$T_{max}$ (mean): 2.7 h vs. 0.5 h</li><li>$T_{1/2}$ (mean): 5.4 h vs. 3.9 h</li><li>$AUC_{0-24}/AUC_{0-24}$ (s.c./i.v.): 0.99</li></ul> | All survived |
| Goutelle, 2018 [13] | Retrospective PK study | Ertapenem (1 g/24 h, 2 g/24 h, 1 g/12 h) | 31 (s.c. vs. i.v.) | BJI | According to the Monte Carlo simulation, subcutaneous administration was associated with higher PTA values than the i.v. route | |
| Roubaud-Baudron, 2019 [15] | Prospective PK study | Ertapenem, 1 g/24 h | 26 (16 s.c. 10 i.v.) | Patients with >65 years (17 UTI, 5 pneumonia, 5) | <ul><li>$C_0$ (mean): 9 mg/L vs. 12 mg/L</li><li>$C_{0.5}$ (mean): 28 mg/L vs. 194 mg/L</li><li>$C_{2.5}$ (mean): 53 mg/L vs. 103 mg/L</li><li>AUC (mean): 1126.92 mg·h/L vs. 1005.27 mg·h/L</li></ul> | Survival at 15 d: 75% vs. 70% |
| Matzneller, 2020 [16] | Prospective PK study | Temocillin, 2 g | 8 (s.c. vs. i.v.) | Healty volunteers | <ul><li>$C_{max}$ (mean): 100 mg/L vs. 233 mg/L</li><li>$C_{12 h}$ (mean): 50.3 mg/L vs. 29.9 mg/L</li><li>$T_{max}$ (mean): 4.8 h vs. 0.67 h</li><li>$T_{1/2}$ (mean): 6.6 h vs. 5.3 h</li><li>$V_d$ (mean): 0.18 L/kg vs. 0.16 L/kg</li><li>$AUC_{0-12}$ (mean): 818.1 mg·h/L vs. 959.4 mg·h/L</li></ul> | - |
| Kado, 2020 [17] | PK study | benzathine penicillin G, 1.2 MIU | 15 (7 s.c. vs. 8 i.m.) | Healty volunteers | <ul><li>$C_{max}$ (???): 36.3 ng/mL vs. 56.8 ng/mL</li><li>$C_{min}$: 10.5 ng/mL vs. 5.2 ng/mL</li><li>T > 20 ng/mL: 29% vs. 32%</li><li>T > 10 ng/mL: 100% vs. 65%</li></ul> | - |

**Table 1.** *Cont*.

| Author, Publication Year | Study Design | Drug, Dosage | Patients (s.c. vs. Control Group) | Characteristics | PK Measures (s.c. vs. Control Group) | Clinical Outcome |
|---|---|---|---|---|---|---|
| Champoux, 1996 [18] | PK study | Ampicillin, 1 g | 24 (s.c. vs. i.v.) | Healty volunteer (12 young volunteers and 12 old volunteers) | <ul><li>$C_{max}$ (mean):<ul><li>Young: 28 µg/L vs. 49 µg/L</li><li>Old: 33 µg/L vs. 39 µg/L</li></ul></li><li>$T_{max}$ (mean):<ul><li>Young: 45′ vs. 23′</li><li>Old: 49′ vs. 27′</li></ul></li><li>$T_{1/2}$ (mean):<ul><li>Young: 84′ vs. 58′</li><li>Old: 114′ vs. 91′</li></ul></li><li>AUCsc/AUCev:<ul><li>Young: 0.92</li><li>Old: 0.96</li></ul></li></ul> | - |

**Table 2.** PK measures reported in the studies included in the review.

| Author, Publication Year | Drug, Dosage (s.c. vs. SoC) | $C_{min}$ (s.c. vs. SoC) | $C_{max}$ (s.c. vs. SoC) | AUC (s.c. vs. SoC or s.c./SoC) |
|---|---|---|---|---|
| Pilmis, 2020 [9] | Cefepime, 3.1 g/day (12 s.c. vs. 12 i.v.) | $C_{min}$ (median): 29.1 mg/L vs. 31.9 mg/L | // | // |
| Walker, 2005 [8] | Cefepime, 1 g (10 s.c. vs. i.v.) | // | // | $AUC_{0-\infty}$ (mean): 134.8 h·mg/L vs. 137 h·mg/L |
| Borner, 1985 [5] | Ceftriaxone (0.5 g s.c. vs. 0.5 g i.v. vs. 2 g i.v) | $C_{min}$ (mean) at 24 h: 6.6 mg/L vs. 6.5 mg/L | $C_{max}$ (mean): 37.1 mg/L vs. 83.8 mg/L | AUCsc/AUCev: 0.96 |
| Harb, 2010 [7] | Ceftriaxone, 1 g (29 ialuronidase-facilated s.c. infusion vs. standard s.c. infusion vs. i.v. infusion) | // | $C_{max}$ (mean): 92 µg/mL vs. 82.2 µg/mL vs. 150 µg/mL | $AUC_{0-t}$ (mean): 1139.3 µg·h/L vs. 1115.6 µg·h/L vs. 1065 µg·h/L |

**Table 2.** *Cont.*

| Author, Publication Year | Drug, Dosage (s.c. vs. SoC) | $C_{min}$ (s.c. vs. SoC) | $C_{max}$ (s.c. vs. SoC) | AUC (s.c. vs. SoC or s.c./SoC) |
|---|---|---|---|---|
| Ebihara, 2016 [6] | Ceftazidime, 0.5 g/10 mL in continuous infusion (without mentholate compression vs. mentholate compression) | // | $C_{max}$: 44.8 µg/mL vs. 57.4 µg/mL | // |
| Frasca, 2009 [12] | Ertapenem, 1 g/24 h (6 s.c. vs. i.v.) | | $C_{max}$ (mean): 43 µg/mL vs. 115 µg/mL | $AUC_{0-24}/AUC_{0-24}$ (s.c./i.v.): 0.99 |
| Roubaud-Baudron, 2019 [15] | Ertapenem, 1 g/24 h, (10 i.v. vs. 16 s.c.) | | | AUC (mean): 1126.92 mg·h/L vs. 1005.27 mg·h/L |
| Matzneller, 2020 [16] | Temocillin, 2 g (8 s.c. vs. i.v.) | $C_{12\,h}$ (mean): 50.3 mg/L vs. 29.9 mg/L | $C_{max}$ (mean): 100 mg/L vs. 233 mg/L | $AUC_{0-12}$ (mean): 818.1 mg·h/L vs. 959.4 mg·h/L |
| Kado, 2020 [17] | benzathine penicillin G, 1.2 MIU (7 s.c. vs. 8 i.m.) | $C_{min}$: 10.5 ng/mL vs. 5.2 ng/mL | $C_{max}$: 36.3 ng/mL vs. 56.8 ng/mL | // |
| Champoux, 1996 [18] | Ampicillin, 1 g (24 young s.c. vs. i.v.) | $C_{max}$ (mean):<br>○ Young: 28 µg/L vs. 49 µg/L<br>○ Old: 33 µg/L vs. 39 µg/L | // | AUCsc/AUCev:<br>○ Young: 0.92<br>○ Old: 0.96 |

## 5. Discussion

In this paper, we presented a case of the successful treatment of a patient with nosocomial pneumonia using s.c. piperacillin/tazobactam. In our case report, the piperacillin/tazobactam was administered with a continuous infusion, administering one dose of 4.5 g every six hours for a six-hour period. Piperacillin/tazobactam is stable, using this mode of administration, for 6 h and, according to the literature, it is stable for 24 h [21].

In the case presented in this study, piperacillin blood concentrations reached 52.3 mg/L 4 days after the commencement of subcutaneous administration, 93.73 mg/L at 7 days, and 156.39 mg/L 10 days after the start of antibiotic therapy. Unfortunately, in this particular case, we did not achieve microbiological isolation, which precluded us from conducting a precise assessment of the $\int$ T/MIC. Nevertheless, when considering one of the worst-case scenarios, 16 mg/L, corresponding to the ECOFF of Pseudomonas aeruginosa, according to the European Committee on Antimicrobial Susceptibility Testing (EUCAST), as a surrogate MIC value [22,23], we observed that, in the initial measurement, piperacillin blood concentrations exceeded the presumed MIC for the suspected bacterium but fell slightly below the PD target of four times the MIC (64 mg/L) specified for beta-lactams [24,25]. However, 7 days and 10 days from the initiation of antibiotic treatment, piperacillin blood concentrations were substantially superior to the PD target, by four times compared to the MIC.

In the existing literature, there is a lack of studies examining the s.c. administration of piperacillin/tazobactam. We identified a solitary study by Kobayashi, which investigated the s.c. administration of piperacillin/tazobactam. Although this study was excluded from our review due to its publication format as a letter to the editor, it reported a similar in-hospital mortality, attributable to the infectious diseases in the s.c. group compared to the i.v. group. However, it also showed a notably higher all-cause mortality among patients receiving piperacillin/tazobactam via the s.c. route [26].

In the comprehensive literature review conducted by us, we aim to provide an overview of the clinical evidence regarding the s.c. administration of beta-lactam antibiotics. Our investigation found several PK and observational studies related to the cephalosporins, penicillins, and carbapenems administered via the s.c. route. The data extracted from these studies collectively reveal that s.c. administration consistently resulted in a delayed and reduced peak blood concentration of these antibiotics when compared to i.v. or i.m. routes. However, it is noteworthy that the $C_{min}$ remains relatively unchanged in comparison to i.v. or i.m. routes. As beta-lactam antibiotics exhibit a time-dependent PK/PD index, the data regarding s.c. administration are intriguing and promising. On the other hand, the s.c. route of administration may be less suitable for concentration-dependent drugs such as aminoglycosides and daptomycin, given the significantly reduced $C_{max}$, as previously discussed in two earlier reviews on this topic [27,28].

The prolonged $t_{1/2}$ associated with the s.c. administration of beta-lactams and the potential for patient self-medication make this route of administration very attractive, especially for infections requiring long-term antibiotic therapy, such as prosthetic joint infections. In our review, we included two case series describing the use of beta-lactams (ceftriaxone, ceftazidime, ertapenem) administered via the s.c. route for the treatment of chronic osteoarticular infections [11,14]. The s.c. administration of beta-lactams could be particularly useful for the treatment of chronic infections caused by Gram-negative bacteria, which are difficult to treat without inducing susceptibility to oral antibiotics, a very common situation in countries where MDR Gram-negative bacteria are frequently isolated. This strategy could be an alternative to the administration of i.v. antibiotics in a hospital setting for the entire duration of the therapy. It is important to underline that, before the acquisition of more robust clinical results on the s.c. administration of antibiotics, it might be helpful to have the possibility of using t.d.m. in order to reach an effective PK/PD target. Without this information, s.c. administration may fail, as reported by Goutelle et al. [14]. Furthermore, proper patient education regarding drug administration is essential, as illustrated by the case of skin necrosis described by Pouderoux [11]. Further

studies exploring the safety and efficacy of subcutaneously administered beta-lactams in this context are warranted.

Regarding the adverse events, the majority of the studies included in the review documented a low incidence of adverse events associated with the s.c. administration of beta-lactams, with most of these events being localized to the injection site. In their prospective investigation on the tolerance of subcutaneously administered antibiotics, Roubaud-Boudron et al. documented an adverse event rate of 21.5% for ceftriaxone and 23.3% for ertapenem. The most frequently reported events were localized pain and induration at the site of injection. The logistic regression analysis conducted by the authors evidenced the use of a rigid catheter for s.c. administration as a factor that is independently associated with adverse events [29]. Enkel et al. conducted a qualitative assessment study using the participants in the SCIP trial [17], revealing a high level of acceptance for the s.c. administration of penicillin G, with the majority of the interviewed individuals reporting only tolerable discomfort [30].

In light of our research, we can conclude that s.c. administration represents a safe and potentially effective alternative to the utilization of beta-lactams. This method appears to have intriguing implications for the pharmacokinetic profiles of these drugs. Both the findings from the literature review and our case report have consistently shown an extension of the antibiotics' half-life ($t_{1/2}$), a characteristic that may hold promise for shortening the length of stay of patients requiring parenteral antibiotic therapy in the future. However, this literature review highlighted the limited availability of data on the s.c. administration of beta-lactam antibiotics in humans. The paucity of randomized comparative studies and the relatively small sample sizes make it challenging to draw definitive conclusions regarding the feasibility of subcutaneous administration for all beta-lactam antibiotics.

Additional pharmacokinetic studies are imperative to gain a more comprehensive understanding of the achievement of pharmacodynamic targets and the clinical effectiveness of subcutaneously administered beta-lactam antibiotics in human subjects.

**Supplementary Materials:** The following supporting information can be downloaded at: https://www.mdpi.com/article/10.3390/idr16010007/s1.

**Author Contributions:** Study concept and design, G.M.L. and C.T.; article screening, full-text review, bias assessment, and data extraction, G.M.L., B.L. and C.T.; interpretation of data, G.M.L., C.T., S.G., R.M., J.A., C.M. and L.M.; drafting the manuscript, G.M.L., B.L., C.M., L.M. and S.G.; critical revision of the manuscript, C.T., R.M. and S.G.; study supervision, C.T., T.C. and R.M. All authors have read and agreed to the published version of the manuscript.

**Funding:** This research received no external funding.

**Institutional Review Board Statement:** Not applicable.

**Informed Consent Statement:** Written informed consent has been obtained from the patient to publish this paper.

**Data Availability Statement:** No new data were generated.

**Conflicts of Interest:** C.T. has received funds for speaking at a symposia organised on behalf of Pfizer, Novartis, Merck Gilead, Zambon, Infectopharm, Shionogy, Menarini, Angelini and Astellas. The other authors declare no conflicts of interest.

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
