# Peer review of "The Subcutaneous Administration of Beta-Lactams: A Case Report and Literary Review—To Do Small Things in a Great Way"

_2036-7449, doi:10.3390/idr16010007_

Round 1

Reviewer 1 Report

Comments and Suggestions for Authors

In my opinion this is a well written, informative review on subcutaneous administration of beta-lactams.

To improve clarity, authors should add details on the s.c. administration route used in their case report. Was an elastomeric pump used? which kind of elastomeric pump? which infusion rate? 2 ml/hour?

Authors should expand the discussion paragraph adding considerations on the possibility to treat (early) discharged patients via elastomeric pump antimicrobial administration - which kind of patient may be eligible for such approach?

Authors should also add considerations on the stability of the mentioned beta-lactams at room/body temperature (i.e., considering the use of refrigerated bags for the elastomeric pump antimicrobial administration?)

Comments on the Quality of English Language

English language in the manuscript is fine and the text is well comprehensible. Only minor editing of English language is required.

Author Response

To improve clarity, authors should add details on the s.c. administration route used in their case report. Was an elastomeric pump used? which kind of elastomeric pump? which infusion rate? 2 ml/hour?

  • Dear reviewer, thank you for your comment. Since the patient was admitted under regular care, the infusion of piperacillin/tazobactam was administered through a normal intravenous line and not through an elastomeric device. However, we have modified the text for a better comprehension (lines 68-74)

Authors should expand the discussion paragraph adding considerations on the possibility to treat (early) discharged patients via elastomeric pump antimicrobial administration - which kind of patient may be eligible for such approach?

  • Done (lines 218-232)

Authors should also add considerations on the stability of the mentioned beta-lactams at room/body temperature (i.e., considering the use of refrigerated bags for the elastomeric pump antimicrobial administration?)

  • Thank you for your comment. There are not reports on the use of elastomeric pump for the s.c. in the current literature. The beta-lactams administered s.c. included in the review were administered rapidly, in fact the longer period of administration reported for beta-lactams was 45’ (Goutelle, 2021). Instead in the case described here piperacillin/tazobactam was administered s.c. continuously and the stability of piperacillin/tazobactam was reported in 188-191

Reviewer 2 Report

Comments and Suggestions for Authors

The authors of the present manuscript describe a case study of an elderly patient with pneumonia (possibly acquired in the hospital) who started IV antibiotic treatment.  However, the IV site did not remain patent and another viable site for the IV catheter could not be found.  The patient was started on a 7-day SC treatment with piperacillin/tazobactam.  The infusion site was massaged every 6 hours (with treatment stopped during this period) and the SC site was changed to the patient's other thigh every 24 hr.  The authors include therapeutic (levels) drug monitoring (TDM).

After this, the authors conducted a literature review of prior clinical studies examining SC beta-lactam treatment.  Standard literature databases were queried, with manuscripts that were not independent research studies being excluded.  A dozen or so manuscripts met criteria for inclusion.  The findings were reviewed with a table of parameters regarding clinical picture (healthy vs infected) including demographics, drug, doses, as well as PK and TDM data.  

I believe the paper provides an important addition to the literature, with the authors recognizing limitations of the study and the literature as well as suggestions for future directions.

I only found a few grammatical errors as follows

Line 50 change "after few days" to "after a few days"

Line 87 change "august 2023" to "August 2023"

Line 118 delete "absolutely"

Line 165 change "We have found" to "We found"

There are a few Table formatting errors.

Author Response

Line 50 change "after few days" to "after a few days"

  • Done

Line 87 change "august 2023" to "August 2023"

  • Done

Line 118 delete "absolutely"

  • done

Line 165 change "We have found" to "We found"

  • done

There are a few Table formatting errors.

  •  We corrected the errors.